# The Best of N Worlds: Aligning Reinforcement Learning with Best-of-N Sampling via max$@k$ Optimization

## Abstract

The application of Reinforcement Learning with Verifiable Rewards (RLVR) to mathematical and coding domains has demonstrated significant improvements in the reasoning and problem-solving abilities of Large Language Models. Despite its success in single generation problem solving, the reinforcement learning fine-tuning process may harm the model's exploration ability, as reflected in decreased diversity of generations and a resulting degradation of performance during Best-of-N sampling for large N values. In this work, we focus on optimizing the max$@k$ metric, a continuous generalization of pass$@k$. We extend on-policy gradient estimate to the off-policy updates, a common element in modern RLVR algorithms, that allows better sample efficiency. Empirically, we show that our objective effectively optimizes max$@k$ metric in off-policy scenarios, aligning the model with the Best-of-N inference strategy.

## 1 Introduction

The current success of Large Language Models (LLMs) (Radford et al., 2019) significantly relies on the application of Reinforcement Learning (RL) during the model post-training process. Typically, LLM training consists of three stages (Liu et al., 2024; Dubey et al., 2024; Hui et al., 2024; Team et al., 2025): unsupervised pre-training on large web corpora, Supervised Fine-Tuning (SFT) on the instruction-following data, and finally the RL stage. The RL stage can address such qualities of the model as helpfulness, instruction following, and safety, which are typically defined by human preferences via the reward model learned from labeled data (Ouyang et al., 2022). In contrast to human preference, coding and math tasks can be evaluated using a verifier, e.g., unit-tests for code tasks. Such a setting is called Reinforcement learning with verifiable rewards (RLVR) (Lambert et al., 2024).

The application of RLVR shows a significant improvement in performance in the single generation setting (DeepSeek-AI et al., 2025). However, the availability of the verifier enables more complex sampling approaches such as Best-of-N (BoN) sampling with further selection via verification. BoN is a simple inference-time scaling approach that generates multiple solutions and chooses the best one based on the response of the verifier (Brown et al., 2024; Chow et al., 2024b). A similar approach is used with the reward model instead of a verifier (Nakano et al., 2021; Gao et al., 2023). The selection of the Best-of-N generations naturally leads to the pass$@k$[1] metric for evaluation.

In a typical RLVR setup, models are trained to directly optimize feedback from a reward model (DeepSeek-AI et al., 2025; Shao et al., 2024; Golubev et al., 2025). While effective in improvement of pass$@1$, this objective can lead to decreased diversity and degraded pass$@k$ scores (Yue et al., 2025; Cui et al., 2025). To mitigate this issue, recent works propose estimators for pass$@k$ objective to directly maximize it (Tang et al., 2025; Chow et al., 2024a; Chen et al., 2025; Walder & Karkhanis, 2025). Such objectives are referred to in the literature as inference-aware ones and they differ in the advantage calculation. Tang et al. (2025); Chow et al. (2024a) assign rewards only to the best completions, while Walder & Karkhanis (2025); Chen et al. (2025) propose different reward transformations that yield an estimate of pass$@k$. However, these methods are designed for

---

[1]We use $k$ and $N$ interchangeably in this work

on-policy RL, whereas state-of-the-art RLVR methods typically combine on-policy and off-policy updates (Schulman et al., 2017; Shao et al., 2024). Moreover, most pass@$k$ estimators formulate the optimization objective with binary rewards, which are sparse and challenging to learn from, limiting stability and sample efficiency (Razin et al., 2025).

Following the setup from the work by Wang et al. (2025), we choose CodeContests (Li et al., 2022) as a source of train and evaluation datasets and LiveCodeBench (Jain et al., 2024), LiveBench (White et al., 2024), CodeForces (Penedo et al., 2025), and MBPP (Austin et al., 2021) as additional evaluation datasets. We focus on coding tasks with verification via test execution, as they naturally provide a continuous reward as the ratio of passed tests. We investigate the dependency of the models' performance after application of different inference-aware RL methods, by measuring pass@$k$ performance at different $k$ values. Furthermore, in the current work, we consider both on-policy and off-policy RLVR.

Our contributions are:

- We show that optimization of continuous reward might be crucial for successful RLVR application.
- We show empirically the effect of RLVR on finetuned models as it decreases diversity while increasing the confidence of its generations.
- We derive an unbiased estimate of max@$k$—the generalization of pass@$k$ for continuous rewards—gradients for off-policy updates. This aligns the fine-tuning process with the inference strategy applied at the test time.
- We show that the proposed objective can efficiently optimize max@$k$ for higher k's while boosting lower k's as well, giving up to $+3.7$ p.p. increase in max@$k$ compared to baselines.

## 2 BACKGROUND

### 2.1 MOTIVATION

**Pass@k Degradation.** To illustrate the problem, we start with a comparison of two models: one obtained just after the instruct finetuning stage and the same model additionally fine-tuned with RLVR on a dataset of coding problems. As a base model, we use Qwen2.5-Coder-7B-Instruct. The fine-tuned version is obtained by application of GRPO (Shao et al., 2024) on CodeContests training dataset (Li et al., 2022) (see Section 4.1 for a complete description of the dataset). As can be seen in Figure 1, RLVR fine-tuning significantly increases the model's pass@$k$ for lower values of $k$. This can be explained by an increase in the model's certainty for some set of examples that the model was not confident in. However, the metric exhibits a decline for higher values of $k$, indicating that a range of correct generations with low probability mass in the base model becomes almost unattainable after application of RL.

To further investigate the underlying differences between the fine-tuned and base models, we consider the distributions of the generations entropy. As can be seen from Figure 2, the RL fine-tuning process skews the distribution of entropy toward zero, resulting in more confident generations and decreased diversity of the samples.

| Setting | Code Contests | | LiveCodeBench | |
|---|---|---|---|---|
| | pass@1 | pass@128 | pass@1 | pass@128 |
| Base model | 0.211 | 0.541 | 0.211 | 0.510 |
| Continuous reward | 0.257 | 0.516 | 0.268 | 0.493 |
| Binary reward | 0.092 | 0.227 | 0.117 | 0.230 |

Table 1: Results of optimization of binary reward and continuous reward on coding tasks.

**Binary vs Continuous Reward.** To highlight the importance of continuous rewards, we compare two training setups that differ in their reward formulation: *binary* and *continuous*. In the binary

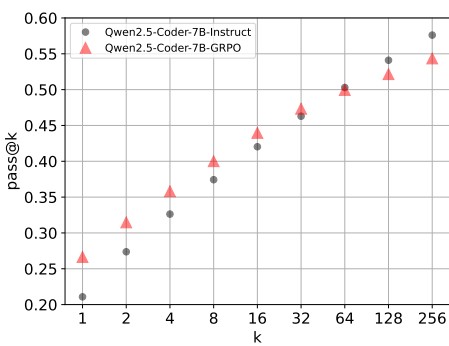

Figure 1: Qwen2.5-Coder-7B-Instruct performance on CodeContests dataset before and after GRPO fine-tuning.

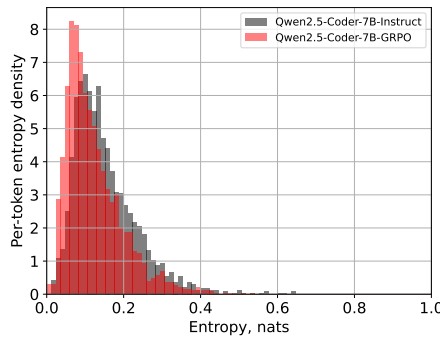

Figure 2: Distribution of the entropy before and after GRPO fine-tuning.

setup, a completion receives a reward of $1$ if and only if it passes all tests; in the continuous setup, the reward is proportional to the fraction of tests passed. Both setups are trained using vanilla GRPO. Table 1 reports the pass@$k$ results for models trained under these two reward schemes, as well as the base model. Comparing the base model with one trained on binary rewards reveals that optimizing for the binary objective (equivalent to the pass@1 metric) actually degrades the model's coding ability. In contrast, optimizing with continuous rewards improves pass@1, but slightly reduces pass@128, which, as mentioned above, might be attributed to decreased diversity in the generations. Those observations motivate our choice of max@$k$ as the target metric to optimize.

## 2.2 PROBLEM STATEMENT

Given a dataset of prompts $D = \{x_i\}_{i=1}^{|D|}$ and a language model $\pi_\theta(y|x)$ parametrized by $\theta$ that defines the policy, a general RL objective to maximize can be written as follows:

$$\mathbb{E}_{x \sim D} \mathbb{E}_{y \sim \pi_\theta(\cdot|x)} r(x, y), \tag{1}$$

where $r(x, y)$ is the reward function. In the language modeling domain, the reward function is typically designed to reflect human preferences (Christiano et al., 2017; Ziegler et al., 2019b). Along with the human preferences, a range of more formally defined rewards is present in the literature spanning mathematical (Hendrycks et al., 2021; Cobbe et al., 2021) and coding domains (Chen et al., 2021; Austin et al., 2021). In this work, we focus on the coding domain, as it provides a continuous reward function, which results in more efficient optimization.

In the coding domain, the reward is typically defined by a set of unit-tests. It can be either binary (whether all tests passed) or continuous (number of tests passed). Given access to these tests, it is possible to generate $k > 1$ solutions and run tests for each one to finally select the best solution at test time. For the scenario with binary reward, Chen et al. (2021) introduced pass@$k$ metric which is the probability of at least one generation out of $k$ being correct (eq. (2) left). In case of continuous reward, max@$k$ metric is used, i.e. maximum reward over subset of size $k$ (eq. (2) right):

$$\text{pass@}k = \mathbb{P}\left(\bigvee_{i=1}^{k}(r(y_i, x) = 1)\right) \qquad \text{max@}k = \mathbb{E}\left(\max\left(\{r(y_i, x)\}_{i=1}^{k}\right)\right). \tag{2}$$

In this study, we consider a general form inference-time strategy where the reward becomes a function of $k$ arguments and eq. (1) can be rewritten as follows:

$$\mathbb{E}_{x \sim D} \mathbb{E}_{y_1 \sim \pi_\theta(\cdot|x), \dots y_k \sim \pi_\theta(\cdot|x)} f(r(y_1, x), \dots r(y_k, x)), \tag{3}$$

where $f$ is an aggregation function that selects the best generation $y$. When $f = \max$, the expectation above becomes the expectation of the max@$k$ metric. Moreover, when $r$ is a binary reward it further transforms into pass@$k$ expectation. In this work, we focus on the derivation of unbiased gradient estimates of the eq. (3) to effectively optimize the model's problem-solving abilities in the case of $k$, when generations are possible.

## 2.3 RELATED WORK

**RL for LLM.**   The policy gradient family of RL algorithms became standard in LLM post-training. Following  (Ranzato et al., 2016; Ziegler et al., 2019a; Stiennon et al., 2020), the key idea of these approaches is to consider the language model as a policy and train it with an RL algorithm. Such RL algorithms include classic Proximal Policy Optimization (PPO) (Schulman et al., 2017), which uses an actor-critic setup and a clipped objective for off-policy updates; reward-free and critic-free methods that directly optimize preferences like Direct Preference Optimization (DPO) (Rafailov et al., 2023) and similar works (Azar et al., 2024; Ethayarajh et al., 2024); and recent robust and effective critic-free algorithms GRPO (Shao et al., 2024), RLOO (Ahmadian et al., 2024), Reinforce++ (Hu et al., 2025), and DAPO (Yu et al., 2025).

**Pass@k after RL.**   Yue et al. (2025) investigate the effect of RLVR finetuning on math, coding, and visual reasoning benchmarks. They show that while RLVR improves pass@1 performance, the base model achieves higher pass@$k$ scores for larger $k$. Furthermore, they observe that RLVR often narrows the boundary of reasoning capabilities. Similarly, Cui et al. (2025) demonstrate a trade-off between model performance and policy entropy: gains in pass@1 performance typically come at the expense of generation diversity, leading to reduced pass@$k$. Subsequent studies report comparable behavior of RLVR, attributing it either to training dataset diversity (Liang et al., 2025) or to the updates on positive sample during training (Zhu et al., 2025).

**Best-of-N sampling.**   Best-of-N (BoN) sampling is a popular alternative to RL finetuning (Stiennon et al., 2020). In BoN, the model generates N completions and the completion with the highest score is returned. Despite its simplicity, BoN has performance comparable to the RL approaches (Yue et al., 2025; Mudgal et al., 2023). Moreover, under some assumptions, BoN is asymptotically equivalent to an optimal policy optimized by RL-constrained KL (Yang et al., 2024; Rafailov et al., 2023). However, BoN can be too computationally expensive, and several works addressed this issue. A range of works address this limitation by fine-tuning the model to directly mimic the BoN distribution: (Sessa et al., 2024) minimize Jeffrey's divergence; (Amini et al., 2024) estimate the cumulative distribution of rewards under BoN and update the policy according to it; (Gui et al., 2024) and (Pace et al., 2024) employ ideas similar to IPO (Azar et al., 2024) and construct their objectives based on best and worst generations. This line of work is orthogonal to the present study since their inference strategy assumes generation of a single completion, as opposed to the generation of $k$ completions employed here.

**Inference-aware finetuning.**   Another line of work explores optimizing models to maximize performance while accounting for inference-time techniques, like BoN. (Chow et al., 2024a) proposes to assign reward only to the best completions during each step and derive a more precise objective for the case of binary rewards. (Tang et al., 2025) explores the idea further and proposes an objective for majority voting and additional variance reduction methods for both BoN and majority voting. In (Walder & Karkhanis, 2025), the authors derive an unbiased estimate for a BoN objective for both binary and continuous rewards and propose an adaptation of leave-one-out baseline for these cases. (Chen et al., 2025) explore binary rewards further and suggests several new unbiased estimates for BoN objective. All these approaches address only the on-policy scenario, while we do not limit ourselves to it and consider both on-policy and off-policy scenarios.

## 3 PASS@K OPTIMIZATION

In this section, we provide derivation of the unbiased gradient estimates of the objective in eq. (3). We derive estimates for two cases: on-policy and off-policy.

To improve readability, we use the following vector notations throughout this section:

$$y_{1:k} \sim \pi^{\otimes k} = \boldsymbol{y} \sim \boldsymbol{\pi}; \quad f\left(r(y_1, x), r(y_2, x), \ldots, r(y_k, x)\right) = f(\boldsymbol{r}(\boldsymbol{y}, x)).$$

## 3.1 PRELIMINARIES

In this work, we mainly use GRPO (Shao et al., 2024) as the optimization method. GRPO is a critic-free RL algorithm that optimizes the policy $\pi_\theta$ by maximizing the following objective:

$$\mathcal{J}_{\text{grpo}} = \mathbb{E}_{x \sim D} \, \mathbb{E}_{\boldsymbol{y} \sim \boldsymbol{\pi}_{\text{old}}(\cdot|x)} \left[ \frac{1}{n} \sum_{i=1}^{n} \left( \mathcal{A}(y_i, x, \varepsilon) - \beta \, \mathbb{D}_{\text{KL}}(\pi_\theta \, \| \, \pi_{\text{old}}) \right) \right],$$

$$\mathcal{A}(y_i, x, \varepsilon) = \min \left( \frac{\pi_\theta(y_i \mid x)}{\pi_{\text{old}}(y_i \mid x)} A_i, \, \text{clip}\left( \frac{\pi_\theta(y_i \mid x)}{\pi_{\text{old}}(y_i \mid x)}, \, 1 - \varepsilon, \, 1 + \varepsilon \right) A_i \right). \tag{4}$$

Here $\pi_{\text{old}}$ is the previous state of optimized policy $\pi_\theta$ from which completions were sampled; $y_i$ completions sampled for the problem $x$ from $\pi_{\text{old}}$; $n$ is the number of sampled completions; $\mathbb{D}_{\text{KL}}$ is a KL-divergence between two distributions; $\beta$ and $\varepsilon$ are hyperparameters; and $A_i$ is an advantage function computed as z-score of completions' rewards $r(y_i, x)$:

$$A_i = \frac{r(y_i, x) - \text{mean}\left( r(y_1, x), r(y_2, x), \dots, r(y_n, x) \right)}{\text{std}\left( r(y_1, x), r(y_2, x), \dots, r(y_n, x) \right)}. \tag{5}$$

The application of z-score to rewards is a form of variance reduction techniques. However, several methods considered in our paper employ different techniques. In those particular cases, we change advantage calculation in GRPO in order to follow the original methods.

In case of strictly on-policy updates ($\pi_\theta = \pi_{\text{old}}$), eq. (4) is equivalent to:

$$\mathcal{J}'_{\text{grpo}} = \mathbb{E}_{x \sim D} \, \mathbb{E}_{\boldsymbol{y} \sim \boldsymbol{\pi}_{\text{old}}(\cdot|x)} \left[ \frac{1}{n} \sum_{i=1}^{n} \left( \log \pi_\theta(y_i \mid x) \, A_i - \beta \, \mathbb{D}_{\text{KL}}(\pi_\theta \, \| \, \pi_{\text{old}}) \right) \right].$$

## 3.2 ON-POLICY CASE

Walder & Karkhanis (2025) derived an unbiased estimator for the max@$k$ gradient. In this section, we present an alternative derivation of the same estimator, which provides a clearer foundation for the subsequent off-policy estimation of the max@$k$ objective. We begin by applying the gradient operator to Equation 3:

$$\nabla_\theta \, \mathbb{E}_{x \sim D} \, \mathbb{E}_{\boldsymbol{y} \sim \boldsymbol{\pi}_\theta(\cdot|x)} f(\boldsymbol{r}(\boldsymbol{y}, x)) = \mathbb{E}_{x \sim D} \nabla_\theta \mathbb{E}_{\boldsymbol{y} \sim \boldsymbol{\pi}_\theta(\cdot|x)} f(\boldsymbol{r}(\boldsymbol{y}, x))$$

$$= \mathbb{E}_{x \sim D} \, \nabla_\theta \int \pi(y_1 \mid x) \dots \pi(y_k \mid x) \, f(\boldsymbol{r}(\boldsymbol{y}, x)) \, d\boldsymbol{y}$$

$$= \mathbb{E}_{x \sim D} \int \sum_{i=1}^{k} \pi(y_1 \mid x) \dots \nabla_\theta \pi(y_i \mid x) \dots \pi(y_k \mid x) \, f(\boldsymbol{r}(\boldsymbol{y}, x)) \, d\boldsymbol{y}.$$

Applying the log-derivative trick $\nabla_\theta \, \pi(y_i \mid x) = \pi(y_i \mid x) \, \nabla_\theta \log \pi(y_i \mid x)$ (Williams, 1992) we get:

$$\mathbb{E}_{x \sim D} \int \sum_{i=1}^{k} \pi(y_1 \mid x) \dots \pi(y_k \mid x) \, \nabla_\theta \log \pi(y_i \mid x) \, f(\boldsymbol{r}(\boldsymbol{y}, x)) \, d\boldsymbol{y}$$

$$= \mathbb{E}_{x \sim D} \, \mathbb{E}_{\boldsymbol{y} \sim \boldsymbol{\pi}_\theta(\cdot|x)} \left[ \sum_{i=1}^{k} \nabla_\theta \log \pi(y_i \mid x) \, f(\boldsymbol{r}(\boldsymbol{y}, x)) \right]. \tag{6}$$

The form of the gradient above allows straightforward estimation via Monte Carlo sampling. For this, we replace expectations with sum over the respective samples and divide by their amount. Expectations over $D$ is replaced with sum over dataset divided by its size. Regarding the expectation over $k$ completions, one can sample some number of completions $n$ and replace expectation with summation over all possible subsets with size $k$ divided by their number $\binom{n}{k}$:

$$\frac{1}{|D|} \sum_{x \in D} \frac{1}{\binom{n}{k}} \sum_{\substack{I \subseteq [n] \\ |I|=k}} \left[ f\left( (r(y_j, x))_{j \in I} \right) \sum_{i \in I} \nabla_\theta \log \pi(y_i \mid x) \right].$$

Resulting objective is an unbiased estimate of eq. (6). Next, let's rearrange terms in the sum and rewrite the expression for each $x$:

$$\frac{1}{\binom{n}{k}} \sum_{i=1}^{n} \left( \nabla_\theta \log \pi(y_i \mid x) \sum_{\substack{I \subseteq [n], i \in I \\ |I|=k}} f\Big( (r(y_j, x))_{j \in I} \Big) \right). \tag{7}$$

Now, let's focus on particular setting where $f = \max$. In that case we can rewrite it as follows:

$$\frac{1}{\binom{n}{k}} \sum_{i=1}^{n} \left( \nabla_\theta \log \pi(y_i \mid x) \sum_{j=1}^{n} w_{ij}\, r(y_j, x) \right). \tag{8}$$

where $w_{ij}$ is a number of sets of size $k$ where both $y_i$ and $y_j$ are present and $y_j$ has the highest score in the set. Let us assume that all generations are sorted in ascending order. With this, $w$ can be trivially calculated: $w_{ii}$ is the number of subsets where $i$ is the highest index, and $w_{ij}$ is the number of subsets containing $i$ where $j$ is the highest index (see Appendix F for more details):

$$w_{ii} = \begin{cases} \binom{i-1}{k-1}, & i \geq k \\ 0, & i < k \end{cases} \quad , \qquad w_{ij} = \begin{cases} \binom{j-2}{k-2}, & j \geq k,\ j > i, \\ 0, & \text{otherwise} \end{cases} \quad .$$

Finally, we can rewrite eq. (8) as follows:

$$\sum_{i=1}^{n} \nabla_\theta \log \pi(y_i \mid x)\, \tilde{r}_i, \quad \tilde{r}_i = \sum_{j=1}^{n} \frac{w_{ij}}{\binom{n}{k}}\, r(y_j, x). \tag{9}$$

In this way, the new objective can be viewed as a reward transformation and all variance reduction techniques can be applied to the new reward $\tilde{r}_i$.

## 3.3 OFF-POLICY

In case of off-policy RL, we still want to maximize same expectation Equation 3. However, we do not have access to the samples from policy $\pi_\theta$, but rather from some other policy $\pi_{\text{old}}$. Therefore, expected reward takes the following form:

$$\mathbb{E}_{x \sim D}\, \mathbb{E}_{\boldsymbol{y} \sim \boldsymbol{\pi}_{\text{old}}(\cdot|x)} \left[ \rho(y_1, x) \ldots \rho(y_k, x)\, f(\boldsymbol{r}(\boldsymbol{y}, x)) \right].$$

where $\rho(y_i, x) = \frac{\pi_\theta(y_i|x)}{\pi_{\text{old}}(y_i|x)}$ is a probability ratio. Similarly to on-policy scenario, we can calculate gradient of the objective:

$$\nabla_\theta \int \pi_{\text{old}}(y_1 \mid x) \ldots \pi_{\text{old}}(y_k \mid x)\, \rho(y_1, x) \ldots \rho(y_k, x)\, f(\boldsymbol{r}(\boldsymbol{y}, x))\, d\boldsymbol{y}$$

$$= \int \sum_{i=1}^{k} \pi_{\text{old}}(y_1 \mid x) \ldots \pi_{\text{old}}(y_k \mid x)\, \rho(y_1, x) \ldots \nabla_\theta\, \rho(y_i, x) \ldots \rho(y_k, x)\, f(\boldsymbol{r}(\boldsymbol{y}, x))\, d\boldsymbol{y}.$$

Again, applying log-derivative trick $\nabla_\theta \rho = \rho \cdot \nabla_\theta \log \rho$ with $\nabla_\theta \log \rho = \nabla_\theta \log \pi_\theta$ we get:

$$\int \sum_{i=1}^{k} \pi_{\text{old}}(y_1 \mid x) \ldots \pi_{\text{old}}(y_k \mid x)\, \rho(y_1, x) \ldots \rho(y_k, x)\, \nabla_\theta \log \pi_\theta(y_i \mid x)\, f(\boldsymbol{r}(\boldsymbol{y}, x))\, d\boldsymbol{y}$$

$$= \mathbb{E}_{\boldsymbol{y} \sim \boldsymbol{\pi}_{\text{old}}(\cdot|x)} \left[ \rho(y_1, x) \ldots \rho(y_k, x)\, f(\boldsymbol{r}(\boldsymbol{y}, x)) \sum_{i=1}^{k} \nabla_\theta \log \pi_\theta(y_i \mid x) \right].$$

Similarly to on-policy scenario, we can estimate gradient above with Monte Carlo sampling:

$$\frac{1}{\binom{n}{k}} \sum_{\substack{I \subseteq [n] \\ |I|=k}} \left[ \boldsymbol{\rho}(I)\, f\Big( (r(y_j, x))_{j \in I} \Big) \sum_{i \in I} \nabla_\theta \log \pi(y_i \mid x) \right], \quad \boldsymbol{\rho}(I) = \prod_{j \in I} \rho(y_j, x).$$

Changing the summation terms:

$$\frac{1}{\binom{n}{k}} \sum_{i=1}^{n} \left[ \nabla_\theta \log \pi \left( y_i \mid x \right) \sum_{\substack{I \subseteq [n], i \in I \\ |I|=k}} \boldsymbol{\rho}(I) \, f\left( (r(y_j, x))_{j \in I} \right) \right]. \tag{10}$$

For the case of $f = \max$, estimation above is exponentially hard to compute. However, in case of PPO-style RL algorithms, off-policy data come from recent states of optimized policy. Therefore, probability ratios are usually close to $1$. We will use that assumption to approximate off-policy gradient estimation. Let's denote $\rho(y_i, x) = 1 + \delta_i$ where $\delta_i \approx 0$, then we can approximate eq. (10) to the first order:

$$\frac{1}{\binom{n}{k}} \sum_{i=1}^{n} \left[ \nabla_\theta \log \pi(y_i \mid x) \sum_{\substack{I \subseteq [n], i \in I \\ |I|=k}} \left( 1 + \sum_{j \in I} \delta_j \right) \max\left( (r(y_j, x))_{j \in I} \right) \right].$$

Following the same notation as with on-policy case, we get (for more details see Appendix G):

$$w'_{ii} = \begin{cases} \binom{i-1}{k-1}(1+\delta_i) + \binom{i-2}{k-2} \sum_{l<i} \delta_l, & i \geq k \\ 0, & i < k \end{cases},$$

$$w'_{ij} = \begin{cases} \binom{j-2}{k-2}(1+\delta_i+\delta_j) + \binom{j-3}{k-3} \sum_{l<j, l \neq i} \delta_l, & j \geq k, \, j > i, \\ 0, & \text{otherwise} \end{cases}.$$

Finally, policy gradient takes form:

$$\frac{1}{\binom{n}{k}} \sum_{i=1}^{n} \left( \nabla_\theta \log \pi(y_i \mid x) \sum_{j=1}^{n} w'_{ij} \, r(y_j, x) \right). \tag{11}$$

## 4 EXPERIMENTS

### 4.1 DATASETS

We select several popular coding benchmarks that are covered with tests: CodeContests (Li et al., 2022), LiveCodeBench (Jain et al., 2024), LiveBench (White et al., 2024), MBPP (Austin et al., 2021), and CodeForces (Penedo et al., 2025). Similarly to (Wang et al., 2025) setup, we process CodeContests and retain only tasks with complexity $\leq 3$. Additionally, to accommodate our GPU restrictions, we filter out any tasks that are longer than $512$ tokens. We randomly split CodeContests into train and test subsets, with $2,490$ examples in train and $276$ in test. For LiveCodeBench (Jain et al., 2024), we use version 6. For CodeForces, we filter out any data overlapping with CodeContests and retain only the test subset for evaluation. Lastly, we process tests for MBPP to support input/output format. In our experiments, LiveCodeBench, LiveBench, CodeForces, and MBPP are used only for evaluation.

### 4.2 MODELS

In our experiments, we use Qwen2.5-7B-Coder-Instruct (Hui et al., 2024) as a base model. This is a modern powerful model built for code tasks. We chose its "Instruct" variation because all of the mentioned datasets are for instructed code generation.

### 4.3 BASELINES

We compare with other inference-aware fine-tuning methods, as they are direct alternatives to our approach. From the studies described in section 2.3, we select all baselines that allow the reward to be a continuous function:

- **BoN-max** - samples $k$ completions and assigns the reward only to the completion with the best score. We use two variance reduction methods:

  - **BoN-max-mean** (Chow et al., 2024a) - the same calculation of advantage as in vanilla GRPO.
  - **BoN-max-second** (Tang et al., 2025) - instead of the mean value of the rewards, subtracts the second-best reward.

- **BoN LOO-1** - an unbiased estimation of the on-policy BoN objective eq. (9) with Leave-One-Out–1 variance reduction, that achieved the best performance as reported by Walder & Karkhanis (2025).

Additionally, we propose a small modification to the advantage calculation with the unbiased on-policy objective and use it as another baseline:

- **BoN mean** - same objective as in BoN LOO-1, but the advantage is calculated as in standard GRPO - by subtracting mean value and dividing by standard deviation.

You can find more information on the baselines in Appendix C.

## 4.4 HYPERPARAMETERS

In our work we use standard hyperparametrs for RLVR Wang et al. (2025). We share hyperparameters across all our approaches. This is done to mitigate their effect on models' comparison. During sampling, we generate 8 completions with the following generation parameters: temperature 1.0, top-p 1.0, and max generation tokens 256. We use Adam optimizer (Kingma & Ba, 2014) with learning rate $5 \cdot 10^{-5}$. For RL, we use common hyperparameters $\beta = 0.01$, $\varepsilon = 0.2$ for clipping, and 3 PPO iterations for off-policy updates. Additionally, we clamp absolute values of $\delta$-s to 0.01, to For all experiments we perform one epoch training, additional experiments with extra epochs are provided in Appendix D.

## 4.5 RESULTS

| Method | CC | LCB | LB | CF | MBPP |
|---|---|---|---|---|---|
| Base model | 0.710 | 0.598 | 0.241 | 0.226 | 0.619 |
| BoN-max second | 0.678 | 0.557 | 0.235 | 0.230 | 0.546 |
| BoN-max mean | 0.702 | 0.593 | 0.222 | 0.242 | 0.609 |
| BoN LOO-1 | 0.647 | 0.530 | 0.227 | 0.200 | 0.484 |
| BoN mean | 0.693 | 0.579 | **0.280** | 0.229 | 0.673 |
| Off-policy BoN (ours) | **0.718\*** | **0.616\*** | 0.278 | **0.255\*** | **0.710\*** |

Table 2: $\max@128$ metric for our method along with the proposed baselines. The top row depicts the following datasets: CodeContests (CC), LiveCodeBench (LCB), LiveBench (LB), CodeForces (CF), Mostly Basic Python Programming (MBPP). Numbers in bold indicate the best performance, and underlined numbers the second-best. Bold number with * denote results that are significantly better than the second best on the same dataset according to Wilcoxon signed-rank test.

Table 2 depicts $\max@128$ metric (see Appendix E for computational details) for the considered datasets and baselines. For all datasets except LiveBench, the proposed off-policy BoN approach shows the best performance. On LiveBench, our method performs similarly to BoN mean approach. The highest gain, equal to 3.7 p.p. is observed on MBPP dataset. To verify these gains, we apply the Wilcoxon signed-rank test (Wilcoxon, 1992) to compare the second best performing method with the Off-policy BoN one. As a result, we reject the null hypothesis with the highest $p$-value of 0.039 for all datasets except LiveBench. For LiveBench, with $p = 0.08$, we fail to reject the null hypothesis, though the result approaches significance. Thus, our proposed approach significantly outperforms baselines on four datasets and performs comparatively to the best method on the 5-th dataset. These observations highlight the effectiveness of our method compared to the baselines

considered. Additionally, *BoN mean* outperforms *BoN LOO-1* across all benchmarks. Those methods have the same objective but different variance reduction techniques, which signifies that *mean* is preferable to *LOO-1*.

Furthermore, Table 3 shows $\max@1$ scores in the same setup. In this case, our method shows the best performance in two out of five settings. The rest are shared by Bon-max second, BoN LOO-1, and BoN mean objectives. Moreover, for MBPP, our approach shows the second to the best performance. Taking into account high pass@128 values for our approach, these results show a potentially fruitful exploration/exploitation trade-off introduced by our method.

For complete evaluation results, refer to Appendix H.

| Method | CC | LCB | LB | CF | MBPP |
|---|---|---|---|---|---|
| Base model | 0.317 | 0.266 | 0.158 | 0.061 | 0.070 |
| BoN-max second | **0.394** | 0.315 | 0.173 | 0.077 | 0.090 |
| BoN-max mean | 0.375 | 0.309 | 0.188 | 0.074 | 0.086 |
| BoN LOO-1 | 0.385 | 0.333 | **0.202** | 0.072 | 0.125 |
| BoN mean | 0.389 | 0.338 | 0.170 | 0.078 | **0.160** |
| Off-policy BoN (ours) | 0.370 | **0.338** | 0.184 | **0.080** | 0.146 |

Table 3: $\max@1$ metric for the considered methods. The abbreviations of the dataset name abbreviation from Table 2.

## 5 LIMITATIONS AND FUTURE WORK

The key limitation of the current work is the requirement of sampling multiple completions for each problem. For a large number of completions, sampling might introduce a significant increase in inference wall-clock time. However, for the BoN sampling, each sample is produced independently, and hence, can be done in parallel, reducing the latency. Furthermore, training with any BoN objective requires sampling at least N completions for each task. This reduces the maximum number of prefixes that can fit into a single batch. However, this can be addressed with gradient accumulation techniques. Finally, this work is focused solely on the optimization of $\max@k$, while other aggregation methods, such as majority voting, can be addressed in the math domain.

Promising directions for future research include extending our approach to mathematical reasoning, which requires carefully constructed continuous rewards; exploring dynamic training schedules that transition from objectives such as max@k to stricter metrics like max@1 or vice versa; closing the gap between max@k and max@1 performance by aligning the distribution of a single generation with that of the best among $k$; investigating conditional sampling schemes in which each generation can leverage previous ones for iterative refinement.

## 6 CONCLUSION

In this work, we addressed the problem of decreased BoN sampling quality after application of RLVR for large values of $N$. Furthermore, we showed that continuous reward is crucial for the successful optimization process. To tackle these problems, we proposed to directly optimize $\max@k$ using policy gradient approach. We show that an unbiased gradient estimate of this objective can be obtained with the application of Monte Carlo sampling. We further extended our derivations to the off-policy case. We provided empirical evidence that the proposed estimate can effectively optimize the model for the BoN inference strategy on coding tasks.

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

## A  LLM USAGE

In this work, we use LLMs only for writing purposes: rephrasing some sentences, grammar and syntax checks, and addition/deletion of articles.

## B  REPRODUCIBILITY STATEMENT

In the supplementary materials we provide our anonymized repository with instructions on how to run and evaluate our experiments.

## C  BASELINES

Here we describe baseline approaches used in our work. All of them modify how advantage is calculated in GRPO formulation:

$$\mathcal{J}_{\text{grpo}} = \mathbb{E}_{x \sim D} \, \mathbb{E}_{\boldsymbol{y} \sim \boldsymbol{\pi}_{\text{old}}(\cdot|x)} \left[ \frac{1}{n} \sum_{i=1}^{n} \left( \mathcal{A}(y_i, x, \varepsilon) - \beta \, \mathbb{D}_{\text{KL}}(\pi_\theta \,\|\, \pi_{\text{old}}) \right) \right],$$

$$\mathcal{A}(y_i, x, \varepsilon) = \min \left( \frac{\pi_\theta(y_i \mid x)}{\pi_{\text{old}}(y_i \mid x)} A_i, \, \text{clip} \left( \frac{\pi_\theta(y_i \mid x)}{\pi_{\text{old}}(y_i \mid x)}, \, 1 - \varepsilon, \, 1 + \varepsilon \right) A_i \right).$$

- In **BoN-max mean** (Chow et al., 2024a) advantage is assigned only to the completions with the highest scores and average score of all completions is used as variance reduction. Let's denote $r_m = \max\left(r(y_1, x), r(y_2, x), \ldots, r(y_n, x)\right)$, then:

$$A_i = \begin{cases} r(y_i, x) - \text{mean}\left(r(y_1, x), r(y_2, x), \ldots, r(y_n, x)\right) & , r(y_i, x) = r_m \\ 0 & , otherwise \end{cases}.$$

- In case of **BoN-max second** (Chow et al., 2024a) the advantage is similar, but second best value is used as variance reduction:

$$A_i = \begin{cases} r(y_i, x) - \max_{i \mid r(y_i, x) \neq r_m}\left(r(y_i, x)\right) & , r(y_i, x) = r_m \\ 0 & , otherwise \end{cases}.$$

- **BoN LOO-1**(Tang et al., 2025) uses on-policy estimation eq. (7) of BoN objective, but modifies to decrease its variance. New objective takes the form of:

$$\frac{1}{\binom{n}{k}} \sum_{i=1}^{n} \left( \nabla_\theta \log \pi(y_i \mid x) \sum_{\substack{I \subseteq [n], i \in I \\ |I| = k}} \max\left(\left(r(y_j, x)\right)_{j \in I}\right) - \max\left(\left(r(y_j, x)\right)_{j \in I \setminus i}\right) \right).$$

The sum of the first terms results in eq. (9). The sum of the second terms Walder & Karkhanis (2025) propose to calculate in the following way:

$$term_2 = \frac{k}{n(k-1)} b_i^{(k-1)},$$

where

$$b_i^{(k)} = \frac{1}{\binom{n-1}{k}} \sum_{\substack{j=1 \\ j \neq i}}^{n} \sum_{\substack{I \subseteq [n] \setminus i, j \in I \\ |I| = k}} \max\left((r(y_l, x))_{l \in I}\right),$$

and calculated recursively:

$$b_1^{(k)} = \frac{1}{\binom{n}{k}} \sum_{i=2}^{n} (w_{ii} + (i-2)w_{i,i+1}) r(y_i, x)$$

$$b_{i+1}^{k} = b_i^{(k)} + \frac{1}{\binom{n}{k}} (r(y_i, x) - r(y_{i+1}, x))(w_{ii} + (i-2)w_{i,i+1})$$

- In **BoN mean** baseline we calculate rewards as described in eq. (9). After that, z-score is applied to these rewards as in normal GRPO (eq. (5)).

## D  ADDITIONAL RLVR RESULTS

| Method | LiveCodeBench | | CodeContests | |
|---|---|---|---|---|
| | max@1 | max@128 | max@1 | max@128 |
| Base model | 0.266 | 0.598 | 0.317 | **0.710** |
| RL one epoch | 0.339 | 0.586 | 0.397 | 0.688 |
| RL four epochs | **0.361** | 0.609 | **0.428** | 0.696 |
| RL PPO=3 one epoch | 0.343 | 0.608 | 0.401 | 0.701 |
| RL PPO=3 four epochs | 0.347 | **0.627** | 0.404 | 0.703 |

Table 4: $\max@1$ and $\max@128$ metrics for the pure RLVR.

In this section we provide additional results for pure RLVR with extended training. Table 4 presents results with more epochs and extra PPO steps. As can be seen from the results, training with more epochs consistently improves the metric. The best $\max@1$ is attained by four epochs pure RL training for both datasets. For CodeContests dataset, as expected, we observe degradation of the $\max@128$ metric even with extra training. At the same time, for LiveCodeBench dataset, the best $\max@128$ is attained by the RL with three PPO steps. This might indicate that the problem of diversity after RL training is less pronounced for certain datasets. Nevertheless, the decrease of $\max@k$ metric might still be present for larger $k$ values.

## E  TECHNICAL DETAILS

### E.1  PASS@$k$ AND MAX@$k$ EVALUATION

To evaluate the pass@$k$ and max@$k$ metrics, we sample 256 completions for each prompt. For pass@$k$, we adopt the unbiased estimator from Chen et al. (2021):

$$\text{pass@}k = 1 - \frac{\binom{n-c}{k}}{\binom{n}{k}},$$

where $n$ denotes the total number of completions and $c$ the number of correct ones. Intuitively, the estimator computes the fraction of subsets of size $k$ that contain at least one correct completion, i.e., $\binom{n}{k} - \binom{n-c}{k}$, normalized by the total number of subsets $\binom{n}{k}$. An analogous estimator can be derived

for max@$k$ by averaging the maximum reward within each subset of size $k$. Let $n$ be the number of generated samples, and let $r_1 \leq r_2 \leq \cdots \leq r_n$ denote their rewards. Then max@$k$ can be estimated as:

$$\max@k = \frac{1}{\binom{n}{k}} \sum_{i=k}^{n} \binom{i-1}{k-1} r_i,$$

where the coefficient $\binom{i-1}{k-1}$ counts the number of subsets of size $k$ whose maximum reward is $r_i$. We show our implementation in Listing 1.

Listing 1: Computation of max@k

```python
def max_at_k(scores: List[float], k: int = 1) -> float:
    """Calculate max@k"""

    scores = np.array(scores)
    sorted_scores = np.sort(scores)
    n = len(scores)

    weights = comb(np.arange(n), k - 1) / comb(n, k)
    max_at_k_score = (weights @ sorted_scores).sum()

    return max_at_k_score
```

### E.2    ON-POLICY BON

We provide our implementation of on-policy BoN objective in Listing 2.

Listing 2: On-policy BoN

```python
def bon_scaler(self, rewards, k):
    n, m = rewards.shape  # n prompts, m generations

    # Calculate scale factors for non diagonal elements (w_ij)
    den = binom(m, k) # total number of sets of size k
    scale = binom(torch.arange(1, m + 1) - 2, k - 2) / den
    scale = scale.nan_to_num(0)

    # broadcast scales for each element. here each column is scale
    scale = scale.repeat(m, 1).T
    scale = torch.tril(scale, diagonal=-1)

    # add diagonal elements to scales (w_ii)
    diag_coef = binom(torch.arange(1, m + 1) - 1, k - 1) / den
    diag_coef = diag_coef.nan_to_num(0)
    diag_coef = torch.diag(diag_coef)

    scale = scale + diag_coef

    # convert to same dtype as rewards
    scale = scale.to(rewards.dtype)
    scale = scale.to(rewards.device)

    # calculate bon rewards
    bon_rewards = rewards @ scale
    return bon_rewards, scale
```

### E.3 OFF-POLICY BON

We provide our implementation of off-policy BoN objective in Listing 3.

Listing 3: Off-policy BoN

```python
def bon_scaler_offpolicy(self, rewards, deltas, k):
    n, m = rewards.shape  # n prompts, m generations
    den = binom(m, k)  # number of k-tuples from generations
    # clamping deltas in case of outliers
    deltas = deltas.clamp(min=-self.clamp_delta, max=self.clamp_delta)
    # cumulative sum of all deltas for correction
    cum_deltas = torch.cumsum(deltas, dim=1) - deltas
    # calculating C(j-2, k-2) scale for off-diagonal elements
    scale_1 = binom(torch.arange(1, m + 1) - 2, k - 2) / den
    scale_1 = scale_1.nan_to_num(0)
    # Broadcast scales for each element
    scale_1 = scale_1.repeat(m, 1).T
    scale_1 = torch.tril(scale_1, diagonal=-1)
    scale_1 = scale_1.to(rewards.device)
    # Add diagonal elements to scales
    diag_scale_1 = binom(torch.arange(1, m + 1) - 1, k - 1) / den  # C(j
        -1, k-1)
    diag_scale_1 = diag_scale_1.nan_to_num(0)
    diag_scale_1 = diag_scale_1.to(rewards.device)

    scale_base = scale_1 + diag_scale_1

    # off-policy correction
    # C(j-2, k-2) * (delta_i + delta_j)
    off_diag_term1 = scale_1 * (deltas.view(n, m, 1) + deltas.view(n, 1,
        m))
    # calculating C(j-3, k-3) scale for off-diagonal elements
    scale_2 = binom(torch.arange(1, m + 1) - 3, k - 3) / den
    scale_2 = scale_2.nan_to_num(0)
    scale_2 = scale_2.repeat(m, 1).T
    scale_2 = torch.tril(scale_2, diagonal=-1)
    scale_2 = scale_2.to(rewards.device)
    # C(j-3, k-3) * (cum_delta_j - delta_i)
    off_diag_term2 = scale_2 * (cum_deltas.view(n, m, 1) - deltas.view(n
        , 1, m))

    off_diag = off_diag_term1 + off_diag_term2
    # C(j-2, k-2)
    diag_scale_2 = binom(torch.arange(1, m + 1) - 2, k - 2) / den
    diag_scale_2 = diag_scale_2.nan_to_num(0)
    diag_scale_1 = diag_scale_1.to(rewards.device)
    diag_scale_2 = diag_scale_2.to(rewards.device)

    # C(i-1, k-1) * (1 + delta_i) + C(i-2, k-2) * cum_delta_i
    diag_term = diag_scale_1 * deltas + diag_scale_2 * cum_deltas
    diag_term = torch.diag_embed(diag_term)

    scale_correction = off_diag + diag_term

    weights = scale_correction + scale_base
    weights = weights.to(rewards.dtype)

    policy_gradient_weights = rewards @ weights

    return policy_gradient_weights, weights
```

# F    ON-POLICY DERIVATION

In this section, we derive on-policy BoN objective in greater detail. We will start from the following form of policy gradient with $f = \max$(eq. (12)):

$$\frac{1}{\binom{n}{k}} \sum_{i=1}^{n} \left( \nabla_\theta \log \pi(y_i \mid x) \sum_{\substack{I \subseteq [n], i \in I \\ |I| = k}} \max\left( (r(y_j, x))_{j \in I} \right) \right). \tag{12}$$

Here we can replace sum over all subsets with sum over all possible max values:

$$\frac{1}{\binom{n}{k}} \sum_{i=1}^{n} \left[ \nabla_\theta \log \pi(y_i \mid x) \sum_{j=1}^{n} r(y_j, x) \sum_{\substack{I \subseteq [n], i \in I \\ |I| = k \\ \max(r(y_l, x))_{l \in I} = r(y_j, x)}} 1 \right].$$

We assume that all completions are sorted by their scores in the ascending order. In that case, $\max\left( r(y_l, x) \right)_{l \in I} = r(y_j, x)$ is equivalent to $I \subseteq [j], j \in I$ and $j \geq i$. Given that, we can simplify the summation:

$$\frac{1}{\binom{n}{k}} \sum_{i=1}^{n} \left[ \nabla_\theta \log \pi(y_i \mid x) \sum_{j=1}^{n} r(y_j, x) \sum_{\substack{I \subseteq [j], i, j \in I \\ |I| = k, j \geq i}} 1 \right].$$

We introduce $w_{ij}$ as:

$$w_{ij} = \sum_{\substack{I \subseteq [j], i, j \in I \\ |I| = k}} 1.$$

With this notation, we get the form of policy gradient from eq. (8) Since $w_{ij}$ is the sum of 1-s over subset, $w_{ij}$ is just equal to the size of that subset:

$$w_{ij} = \left| \{ I \subseteq [j] \; : \; |I| = k, \; i \in I, \; j \in I \} \right|.$$

Consider two cases for $j$:

- If $j = i$ we need to find total number of subsets of size $k$ with elements up to $i$ and containing $i$. Therefore $w_{ij} = \binom{i-1}{k-1}$ if $i \geq k$ and 0 otherwise – we are choosing $k-1$ elements to pair with $i$.

- If $j > i$ we similarly need to find total number of subsets of size $k$ with elements up to $j$ and containing both $i$ and $j$:

$$w_{ij} = \begin{cases} \binom{j-2}{k-2}, & j \geq k, j > i, \\ 0, & \text{otherwise} \end{cases},$$

  here we are selecting $k-2$ elements to add to $i$ and $j$.

# G    OFF-POLICY DERIVATION

The approximation of unbiased off-policy derivation takes the form of:

$$\frac{1}{\binom{n}{k}} \sum_{i=1}^{n} \left[ \nabla_\theta \log \pi(y_i \mid x) \sum_{\substack{I \subseteq [n], i \in I \\ |I| = k}} \left( 1 + \sum_{j \in I} \delta_j \right) \max\left( (r(y_j, x))_{j \in I} \right) \right].$$

Similarly to on-policy case, we can replace sum over all subsets with sum over all possible $\max$ values:

$$\frac{1}{\binom{n}{k}} \sum_{i=1}^{n} \left[ \nabla_\theta \log \pi(y_i \mid x) \sum_{j=1}^{n} r(y_j, x) \sum_{\substack{I \subseteq [n], i \in I \\ |I| = k \\ \max(r(y_l, x))_{l \in I} = r(y_j, x)}} \left( 1 + \sum_{l \in I} \delta_l \right) \right].$$

We assume that all completions are sorted their scores in the ascending order. In that case, $\max(r(y_l, x))_{l \in I} = r(y_j, x)$ is equivalent to $I \subseteq [j]$, $j \in I$ and $j \geq i$. Given that, we can simplify the summation:

$$\frac{1}{\binom{n}{k}} \sum_{i=1}^{n} \left[ \nabla_\theta \log \pi(y_i \mid x) \sum_{j=1}^{n} r(y_j, x) \sum_{\substack{I \subseteq [j], i, j \in I \\ |I| = k, j \geq i}} \left( 1 + \sum_{l \in I} \delta_l \right) \right].$$

We can rewrite this similarly to on-policy case with $w'_{ij}$ coefficients:

$$\frac{1}{\binom{n}{k}} \sum_{i=1}^{n} \left( \nabla_\theta \log \pi(y_i \mid x) \sum_{j=i}^{n} w'_{ij} \, r(y_j, x) \right),$$

where

$$w'_{ij} = \sum_{\substack{I \subseteq [j], i, j \in I \\ |I| = k}} \left( 1 + \sum_{l \in I} \delta_l \right).$$

Consider two cases for $j$:

- In case of $j = i$ we get:

$$w'_{ii} = \sum_{\substack{I \subseteq [i], i \in I \\ |I| = k}} \left( 1 + \sum_{l \in I} \delta_l \right).$$

If $i < k$, then the sum is over empty set and is equal to $0$. Otherwise, we can regroup summed deltas. In the sum $1 + \delta_i$ appears in every term, therefore it will counted $\binom{i-1}{k-1}$ times (total number of subsets of size $k$ with $i$ is a maximum element). All other deltas appear in the sum $\binom{i-2}{k-2}$ – number of subsets that contains both $i$ and any particular element before it. Combining everything we get:

$$w'_{ii} = \begin{cases} \binom{i-1}{k-1}(1 + \delta_i) + \binom{i-2}{k-2} \sum_{l < i} \delta_l, & i \geq k \\ 0, & i < k \end{cases}.$$

- In case of $j > i$ we can similarly group deltas in the summation. Since $I$ always contains $i$ and $j$, then $1 + \delta_i + \delta_j$ appears in every term and will be counted $\binom{j-2}{k-2}$ times (total number of subsets of size $k$ with $i$ and $j$ being the maximum element). All other deltas appear in the sum $\binom{j-3}{k-3}$ – number of subsets that contains both $i, j$ and any particular element before them. Combining everything we get:

$$w'_{ij} = \begin{cases} \binom{j-2}{k-2}(1 + \delta_i + \delta_j) + \binom{j-3}{k-3} \sum_{l < j, l \neq i} \delta_l, & j \geq k, j > i, \\ 0, & \text{otherwise} \end{cases}.$$

# H    FULL EVALUATION RESULTS

We present full evaluation results for CodeContests in Table 5 and Table 6; for LiveCodeBench in Table 7 and Table 8; for LiveBench in Table 9 and Table 10; for CodeForces in Table 11 and Table 12; for MBPP in Table 13 and Table 14.

| Method | pass@1 | pass@2 | pass@4 | pass@8 | pass@16 | pass@32 | pass@64 | pass@128 | pass@256 |
|---|---|---|---|---|---|---|---|---|---|
| Base model | 0.211 | 0.274 | 0.326 | 0.374 | 0.420 | 0.463 | 0.503 | 0.541 | 0.576 |
| BoN-max second | 0.261 | 0.302 | 0.338 | 0.376 | 0.415 | 0.450 | 0.482 | 0.511 | 0.536 |
| BoN-max mean | 0.252 | 0.305 | 0.349 | 0.392 | 0.434 | 0.471 | 0.502 | 0.528 | 0.547 |
| BoN LOO-1 | 0.256 | 0.284 | 0.312 | 0.342 | 0.376 | 0.412 | 0.445 | 0.472 | 0.496 |
| BoN mean | 0.260 | 0.312 | 0.357 | 0.401 | 0.441 | 0.475 | 0.501 | 0.522 | 0.543 |
| Off-policy BoN (ours) | 0.248 | 0.303 | 0.348 | 0.392 | 0.436 | 0.479 | 0.519 | 0.553 | 0.587 |

Table 5: pass@$k$ scores on CodeContests for different values of $k$. The darker the color the higher is the score withing its column.

| Method | max@1 | max@2 | max@4 | max@8 | max@16 | max@32 | max@64 | max@128 | max@256 |
|---|---|---|---|---|---|---|---|---|---|
| Base model | 0.317 | 0.411 | 0.485 | 0.544 | 0.595 | 0.637 | 0.675 | 0.710 | 0.739 |
| BoN-max second | 0.394 | 0.458 | 0.509 | 0.554 | 0.594 | 0.627 | 0.654 | 0.678 | 0.699 |
| BoN-max mean | 0.375 | 0.453 | 0.512 | 0.562 | 0.605 | 0.641 | 0.672 | 0.702 | 0.732 |
| BoN LOO-1 | 0.385 | 0.435 | 0.480 | 0.521 | 0.559 | 0.593 | 0.622 | 0.647 | 0.669 |
| BoN mean | 0.389 | 0.465 | 0.522 | 0.569 | 0.608 | 0.640 | 0.668 | 0.693 | 0.716 |
| Off-policy BoN (ours) | 0.370 | 0.450 | 0.512 | 0.565 | 0.613 | 0.654 | 0.689 | 0.718 | 0.742 |

Table 6: max@$k$ scores on CodeContests for different values of $k$. The darker the color the higher is the score withing its column.

| Method | pass@1 | pass@2 | pass@4 | pass@8 | pass@16 | pass@32 | pass@64 | pass@128 | pass@256 |
|---|---|---|---|---|---|---|---|---|---|
| Base model | 0.211 | 0.283 | 0.341 | 0.386 | 0.422 | 0.454 | 0.482 | 0.510 | 0.540 |
| BoN-max second | 0.255 | 0.307 | 0.350 | 0.385 | 0.415 | 0.442 | 0.468 | 0.493 | 0.514 |
| BoN-max mean | 0.248 | 0.311 | 0.358 | 0.395 | 0.427 | 0.456 | 0.485 | 0.517 | 0.548 |
| BoN LOO-1 | 0.262 | 0.304 | 0.339 | 0.367 | 0.392 | 0.416 | 0.439 | 0.458 | 0.475 |
| BoN mean | 0.271 | 0.327 | 0.369 | 0.402 | 0.432 | 0.458 | 0.483 | 0.505 | 0.531 |
| Off-policy BoN (ours) | 0.272 | 0.328 | 0.372 | 0.408 | 0.440 | 0.468 | 0.495 | 0.524 | 0.557 |

Table 7: pass@$k$ scores on LiveCodeBench for different values of $k$. The darker the color the higher is the score withing its column.

| Method | max@1 | max@2 | max@4 | max@8 | max@16 | max@32 | max@64 | max@128 | max@256 |
|---|---|---|---|---|---|---|---|---|---|
| Base model | 0.070 | 0.123 | 0.201 | 0.296 | 0.395 | 0.484 | 0.556 | 0.619 | 0.675 |
| BoN-max second | 0.090 | 0.145 | 0.214 | 0.292 | 0.369 | 0.437 | 0.495 | 0.546 | 0.600 |
| BoN-max mean | 0.086 | 0.145 | 0.224 | 0.316 | 0.406 | 0.486 | 0.551 | 0.609 | 0.669 |
| BoN LOO-1 | 0.125 | 0.185 | 0.247 | 0.305 | 0.357 | 0.403 | 0.443 | 0.484 | 0.536 |
| BoN mean | 0.160 | 0.250 | 0.349 | 0.438 | 0.511 | 0.572 | 0.625 | 0.673 | 0.720 |
| Off-policy BoN (ours) | 0.146 | 0.235 | 0.339 | 0.440 | 0.530 | 0.602 | 0.660 | 0.710 | 0.750 |

Table 14: max@$k$ scores on MBPP for different values of $k$. The darker the color the higher is the score withing its column.

| Method | max@1 | max@2 | max@4 | max@8 | max@16 | max@32 | max@64 | max@128 | max@256 |
|---|---|---|---|---|---|---|---|---|---|
| Base model | 0.266 | 0.349 | 0.412 | 0.459 | 0.497 | 0.531 | 0.564 | 0.598 | 0.638 |
| BoN-max second | 0.315 | 0.371 | 0.416 | 0.452 | 0.482 | 0.507 | 0.532 | 0.557 | 0.580 |
| BoN-max mean | 0.309 | 0.379 | 0.428 | 0.465 | 0.497 | 0.526 | 0.557 | 0.593 | 0.631 |
| BoN LOO-1 | 0.333 | 0.379 | 0.414 | 0.443 | 0.468 | 0.490 | 0.511 | 0.530 | 0.549 |
| BoN mean | 0.338 | 0.400 | 0.442 | 0.474 | 0.501 | 0.527 | 0.552 | 0.579 | 0.609 |
| Off-policy BoN (ours) | 0.338 | 0.398 | 0.443 | 0.479 | 0.512 | 0.543 | 0.575 | 0.616 | 0.654 |

Table 8: max@$k$ scores on LiveCodeBench for different values of $k$. The darker the color the higher is the score withing its column.

| Method | pass@1 | pass@2 | pass@4 | pass@8 | pass@16 | pass@32 | pass@64 | pass@128 | pass@256 |
|---|---|---|---|---|---|---|---|---|---|
| Base model | 0.150 | 0.181 | 0.196 | 0.204 | 0.210 | 0.216 | 0.220 | 0.225 | 0.235 |
| BoN-max second | 0.166 | 0.190 | 0.202 | 0.208 | 0.214 | 0.218 | 0.221 | 0.225 | 0.235 |
| BoN-max mean | 0.182 | 0.196 | 0.204 | 0.209 | 0.214 | 0.215 | 0.216 | 0.216 | 0.216 |
| BoN LOO-1 | 0.195 | 0.200 | 0.203 | 0.208 | 0.213 | 0.215 | 0.216 | 0.216 | 0.216 |
| BoN mean | 0.162 | 0.190 | 0.205 | 0.214 | 0.223 | 0.233 | 0.248 | 0.268 | 0.294 |
| Off-policy BoN (ours) | 0.174 | 0.192 | 0.198 | 0.201 | 0.205 | 0.214 | 0.229 | 0.250 | 0.275 |

Table 9: pass@$k$ scores on LiveBench for different values of $k$. The darker the color the higher is the score withing its column.

| Method | max@1 | max@2 | max@4 | max@8 | max@16 | max@32 | max@64 | max@128 | max@256 |
|---|---|---|---|---|---|---|---|---|---|
| Base model | 0.158 | 0.188 | 0.203 | 0.212 | 0.220 | 0.227 | 0.233 | 0.241 | 0.255 |
| BoN-max second | 0.173 | 0.197 | 0.209 | 0.216 | 0.221 | 0.225 | 0.229 | 0.235 | 0.248 |
| BoN-max mean | 0.188 | 0.203 | 0.210 | 0.216 | 0.220 | 0.222 | 0.222 | 0.222 | 0.222 |
| BoN LOO-1 | 0.202 | 0.207 | 0.210 | 0.215 | 0.220 | 0.223 | 0.225 | 0.227 | 0.229 |
| BoN mean | 0.170 | 0.198 | 0.213 | 0.222 | 0.231 | 0.241 | 0.257 | 0.280 | 0.307 |
| Off-policy BoN (ours) | 0.184 | 0.201 | 0.206 | 0.211 | 0.218 | 0.230 | 0.249 | 0.278 | 0.320 |

Table 10: max@$k$ scores on LiveBench for different values of $k$. The darker the color the higher is the score withing its column.

| Method | pass@1 | pass@2 | pass@4 | pass@8 | pass@16 | pass@32 | pass@64 | pass@128 | pass@256 |
|---|---|---|---|---|---|---|---|---|---|
| Base model | 0.010 | 0.014 | 0.017 | 0.018 | 0.018 | 0.019 | 0.019 | 0.021 | 0.024 |
| BoN-max second | 0.014 | 0.017 | 0.018 | 0.018 | 0.018 | 0.018 | 0.018 | 0.018 | 0.018 |
| BoN-max mean | 0.013 | 0.016 | 0.018 | 0.018 | 0.018 | 0.019 | 0.019 | 0.021 | 0.024 |
| BoN LOO-1 | 0.012 | 0.015 | 0.017 | 0.018 | 0.018 | 0.018 | 0.018 | 0.018 | 0.018 |
| BoN mean | 0.013 | 0.015 | 0.017 | 0.019 | 0.020 | 0.021 | 0.023 | 0.024 | 0.024 |
| Off-policy BoN (ours) | 0.014 | 0.017 | 0.018 | 0.019 | 0.019 | 0.021 | 0.023 | 0.026 | 0.030 |

Table 11: pass@$k$ scores on CodeForces for different values of $k$. The darker the color the higher is the score withing its column.

| Method | max@1 | max@2 | max@4 | max@8 | max@16 | max@32 | max@64 | max@128 | max@256 |
|---|---|---|---|---|---|---|---|---|---|
| Base model | 0.061 | 0.083 | 0.107 | 0.132 | 0.157 | 0.181 | 0.205 | 0.226 | 0.246 |
| BoN-max second | 0.077 | 0.100 | 0.122 | 0.146 | 0.170 | 0.193 | 0.212 | 0.230 | 0.248 |
| BoN-max mean | 0.074 | 0.098 | 0.122 | 0.145 | 0.168 | 0.192 | 0.217 | 0.242 | 0.264 |
| BoN LOO-1 | 0.072 | 0.091 | 0.111 | 0.129 | 0.148 | 0.167 | 0.185 | 0.200 | 0.214 |
| BoN mean | 0.078 | 0.099 | 0.122 | 0.144 | 0.166 | 0.187 | 0.208 | 0.229 | 0.251 |
| Off-policy BoN (ours) | 0.080 | 0.102 | 0.125 | 0.150 | 0.176 | 0.202 | 0.228 | 0.255 | 0.282 |

Table 12: max@$k$ scores on CodeForces for different values of $k$. The darker the color the higher is the score withing its column.

| Method | pass@1 | pass@2 | pass@4 | pass@8 | pass@16 | pass@32 | pass@64 | pass@128 | pass@256 |
|---|---|---|---|---|---|---|---|---|---|
| Base model | 0.062 | 0.109 | 0.177 | 0.260 | 0.347 | 0.426 | 0.494 | 0.554 | 0.606 |
| BoN-max second | 0.080 | 0.128 | 0.188 | 0.255 | 0.323 | 0.385 | 0.439 | 0.487 | 0.534 |
| BoN-max mean | 0.076 | 0.129 | 0.199 | 0.279 | 0.359 | 0.430 | 0.489 | 0.543 | 0.597 |
| BoN LOO-1 | 0.109 | 0.161 | 0.215 | 0.266 | 0.312 | 0.353 | 0.389 | 0.425 | 0.471 |
| BoN mean | 0.142 | 0.223 | 0.310 | 0.389 | 0.457 | 0.515 | 0.566 | 0.613 | 0.661 |
| Off-policy BoN (ours) | 0.128 | 0.206 | 0.296 | 0.386 | 0.468 | 0.537 | 0.596 | 0.649 | 0.692 |

Table 13: pass@$k$ scores on MBPP for different values of $k$. The darker the color the higher is the score withing its column.

# I COMPLEXITY OF OFF-POLICY ESTIMATE

In this section, we analyze the computational complexity of our method. The number of prompts ($n$) is omitted, as it serves only as a linear scaling factor. The core function `bon_scaler_offpolicy` has a time complexity of $\mathbf{O(m^2)}$ (Listing 4). In addition, computing the $\delta$-values contributes another $\mathbf{O(m^2)}$, and sorting the rewards in ascending order introduces an $\mathbf{O(m \log m)}$ cost. Combining these terms, the overall complexity of the method remains $\mathbf{O(m^2)}$.

Listing 4: Off-policy BoN with complexity

```python
def bon_scaler_offpolicy(self, rewards, deltas, k):
    n, m = rewards.shape  O(1)
    den = binom(m, k)  O(1) # since we can precompute it
    deltas = deltas.clamp(...)  O(m)
    cum_deltas = torch.cumsum(deltas, dim=1) - deltas  O(m)
    # code below can be precomputed therefore it is  O(1)
    scale_1 = binom(torch.arange(1, m + 1) - 2, k - 2) / den
    scale_1 = scale_1.nan_to_num(0)
    scale_1 = scale_1.repeat(m, 1).T
    scale_1 = torch.tril(scale_1, diagonal=-1)
    scale_1 = scale_1.to(rewards.device)
    diag_scale_1 = binom(torch.arange(1, m + 1) - 1, k - 1) / den  # C(j
        -1, k-1)
    diag_scale_1 = diag_scale_1.nan_to_num(0)
    diag_scale_1 = diag_scale_1.to(rewards.device)
    scale_base = scale_1 + diag_scale_1

    # off-policy correction  O(m^2)
    off_diag_term1 = scale_1 * (deltas.view(n, m, 1) + deltas.view(n, 1,
         m))
    # code below can be precomputed therefore it is  O(1)
    scale_2 = binom(torch.arange(1, m + 1) - 3, k - 3) / den
    scale_2 = scale_2.nan_to_num(0)
    scale_2 = scale_2.repeat(m, 1).T
    scale_2 = torch.tril(scale_2, diagonal=-1)
    scale_2 = scale_2.to(rewards.device)
    # C(j-3, k-3) * (cum_delta_j - delta_i)  O(m^2)
    off_diag_term2 = scale_2 * (cum_deltas.view(n, m, 1) - deltas.view(n
        , 1, m))

    off_diag = off_diag_term1 + off_diag_term2  O(m^2)
    # code below can be precomputed therefore it is  O(1)
    diag_scale_2 = binom(torch.arange(1, m + 1) - 2, k - 2) / den
    diag_scale_2 = diag_scale_2.nan_to_num(0)
    diag_scale_1 = diag_scale_1.to(rewards.device)
    diag_scale_2 = diag_scale_2.to(rewards.device)

    # C(i-1, k-1) * (1 + delta_i) + C(i-2, k-2) * cum_delta_i  O(m)
    diag_term = diag_scale_1 * deltas + diag_scale_2 * cum_deltas
    diag_term = torch.diag_embed(diag_term)

    scale_correction = off_diag + diag_term  O(m^2)

    weights = scale_correction + scale_base  O(m^2)
    weights = weights.to(rewards.dtype)

    policy_gradient_weights = rewards @ weights  O(m^2)

    return policy_gradient_weights, weights
```

