# OpenReview forum: "The Best of N Worlds: Aligning Reinforcement Learning with Best-of-N Sampling via max@k Optimization"
_ICLR.cc/2026/Conference — Submitted to ICLR 2026_

### Official Review · Reviewer_wYXd · 2025-10-30

**Soundness:** 3
**Presentation:** 3
**Contribution:** 1
**Rating:** 2
**Confidence:** 3

**Summary:**

This paper proposes a method to improve performance of models fine-tuned with RLVR on mathematical reasoning tasks. As many models are used with best-of-N sampling at inference time, the paper proposes to explicitly optimize the max @ k metric which is compatible with the best-of-N sampling technique. It builds on existing estimators for the policy gradient of max @ k reward, and extends them to the off-policy setting prevalent in RLVR methods. The paper presents a few experiments to support the usage of this optimization target.

**Strengths:**

The paper presents its ideas and proposed contributions clearly and simply, the reasoning and motivation are easy to follow into the proposed method and experiments.

The derivation of the policy gradient estimator seems sound and correct.

Using the proposed estimator, it seems like the proposed method outperforms baselines on the targeted metric of max @ k for large k.

**Weaknesses:**

My main concern with this paper is that the proposed method and results seem incremental and slightly insignificant.

As the authors themselves note, the unbiased estimator for the max @ k policy gradient was developed by Walder & Karkhanis. The motivation for deriving this gradient has also been described in previous papers. In addition, while many modern RLVR algorithms are indeed effectively off-policy, most are designed as on-policy algorithms and assumptions are only relaxed due to necessity of implementation. The same holds for the max @ k objective estimator for the on-policy case. Given all of this, it seems like the novelty of this paper is very limited.

This compounds with the results being unimpressive - while found significant by the Wilcoxon test, it seems like there is very little gain over baselines. It is also lacking that results are not reported for k values other.than 128 (eg. similarly to the plot in Figure 1). Clarity in the way the baselines were run and measured would be a significant addition - see some questions below.

Finally, it is worth wondering whether optimizing for best-of-N sampling, which in itself is necessary only because our models are imperfect, is good algorithmic design. It may border on the realm of Goodhart’s law - if we optimize for best-of-N, do we give up on improving top-1 performance?

**Questions:**

1. Figures 1/2: the effects do not seem very pronounced. How has the statistical significance of the motivating results been measured?
2. In the Off-Policy BoN approach (as presented in the experiments), is the policy gradient presented at the end of section 3 evaluated directly, of is it incorporated into GRPO? If it is the latter, how is it incorporated?
3. Following up on Q2, is any variance reduction performed? Does it follow mean or LOO-1 (comparing to the on policy version)?
4. Does the derived policy gradient estimator incur higher computational complexity than standard reward? How does it compare to standard methods in terms of runtime/extra computation?

---

> ### Author Response · Authors · 2025-11-20
> **Response to Reviewer wYXd**
>
> Thank you for time reviewing our paper and providing feedback to it. We tried to address your concerns below.
>
> >As the authors themselves note, the unbiased estimator for the max @ k policy gradient was developed by Walder & Karkhanis. The motivation for deriving this gradient has also been described in previous papers. In addition, while many modern RLVR algorithms are indeed effectively off-policy, most are designed as on-policy algorithms and assumptions are only relaxed due to necessity of implementation. The same holds for the max @ k objective estimator for the on-policy case. Given all of this, it seems like the novelty of this paper is very limited.
>
> While Walder & Karkhanis provide derivation for max@k objective, they don’t provide any real data experiments with it and report only with pass@k results. In our work, we show that their exact approach (with loo-1 regularisation) is suboptimal and doesn’t improve max@k for large k compared to the base model (table 2 row 1 vs row 4). Additionally, we show that a simple change to the variance reduction can improve its performance (bon mean in table 2). And on top of that, switching from on-policy estimate to off-policy estimate improves it further. These results highlight meaningful extensions beyond prior work.
>
> > ..while many modern RLVR algorithms are indeed effectively off-policy, most are designed as on-policy algorithms and assumptions are only relaxed due to necessity of implementation.
>
> Despite the fact that many modern RLVR algorithms are designed as on-policy, they are effectively applied off-policy in practice, so off-policy considerations are essential. Moreover, there is a growing body of work developing truly off-policy methods (e.g., in asynchronous RL  [(Reimer et al. 2025)](https://arxiv.org/abs/2412.14355) and [(Noukhovitch et al. 2025)](https://arxiv.org/abs/2410.18252)).
>
> > Figures 1/2: the effects do not seem very pronounced. How has the statistical significance of the motivating results been measured?
>
> We did additional statistical tests to check significance of the provided results. The results are the following: the median of the perplexity is significantly lower after GRPO fine-tuning according to the Wilcoxon paired test, the differences in pass@1 and pass@256 scores are statistically significant as well according to the same test. Furthermore, [(Yue et al., 2025)](https://arxiv.org/abs/2407.14622) provided similar results showing that RLVR lead to decrease in pass@k for large k values.
>
> > In the Off-Policy BoN approach (as presented in the experiments), is the policy gradient presented at the end of section 3 evaluated directly, of is it incorporated into GRPO? If it is the latter, how is it incorporated?
>
> It is incorporated into GRPO. Specifically, after normal reward calculation, rewards are re-weighted according to equation 11 (in the updated version), and afterwards we proceed with normal GRPO algorithm.
>
> >Following up on Q2, is any variance reduction performed? Does it follow mean or LOO-1 (comparing to the on policy version)?
>
> We apply mean-based variance reduction in our method. In experiments comparing LOO-1 and mean variance reduction using the on-policy estimator, we found that mean consistently performed better across all datasets (see Table 2, lines 4 and 5). Therefore, we adopted mean as our variance reduction strategy.
>
> > Does the derived policy gradient estimator incur higher computational complexity than standard reward? How does it compare to standard methods in terms of runtime/extra computation?
>
> Yes, the derived policy gradient estimator does incur higher computational complexity compared to using a standard reward. It adds an additional $O(m^2)$ complexity to computations. For more information refer to Appendix I.
>
>
> > Finally, it is worth wondering whether optimizing for best-of-N sampling, which in itself is necessary only because our models are imperfect, is good algorithmic design. It may border on the realm of Goodhart’s law - if we optimize for best-of-N, do we give up on improving top-1 performance?
>
> While this question is completely valid, in our experiments we saw an increase of top-1 performance compared to the base model, so improving Best-of-N performance did not harm top-1 performance. However, there are methods to improve pass@ performance that distill the Best-of-N distribution into a single-shot distribution such as [(Sessa et al., 2024)](https://arxiv.org/abs/2407.14622) and [(Amini et al., 2024)](https://arxiv.org/abs/2407.06057). These methods can be applied on top of our one to obtain better single-shot performance. Additionally, training for combination of both best-of-n and top-1 have been shown to jointly improve both performances  [(Walder & Karkhanis 2025)](https://arxiv.org/abs/2505.15201) and [(Chen et al. 2025)](https://arxiv.org/abs/2508.10751).

---

> > ### Author Response · Authors · 2025-11-20
> > **Response to  Reviewer wYXd part 2**
> >
> > > It is also lacking that results are not reported for k values other.than 128 (eg. similarly to the plot in Figure 1).
> >
> > These results can be found in the Appendix H.
> >
> > > Clarity in the way the baselines were run and measured would be a significant addition - see some questions below.
> >
> > On top of details provided here more information on how the baselines were run and measured are provided in the Appendix. We would be happy to clarify further if it is needed.

---

### Official Review · Reviewer_THfK · 2025-11-01

**Soundness:** 2
**Presentation:** 2
**Contribution:** 2
**Rating:** 4
**Confidence:** 3

**Summary:**

This paper addresses the limitation of current Reinforcement Learning with Verifiable Rewards (RLVR) methods, which optimize for single-sample accuracy but often reduce generation diversity, leading to suboptimal performance when applying Best-of-N (BoN) sampling strategies at inference time. Focusing on the code generation task, the authors propose a novel optimization objective based on max@k, a continuous generalization of the pass@k metric. They derive an unbiased gradient estimator for both on-policy and off-policy cases, enabling direct optimization of max@k. Empirical results on several code benchmarks demonstrate that the proposed off-policy BoN objective improves max@k performance over strong inference-aware baselines, showing its effectiveness in aligning RL optimization with BoN inference in code generation tasks.

**Strengths:**

- This paper proposes a novel optimization objective based on max@k which is a continuous generalization of the pass@k metric for aligning RL optimization with BoN inference in continuous reward settings like code generation tasks.
- This paper proposes an unbiased gradient estimator for both on-policy and off-policy cases, enabling direct optimization of max@k.
- The empirical results on several code benchmarks demonstrate that the proposed off-policy BoN objective improves max@k performance over strong inference-aware baselines, showing its effectiveness in aligning RL optimization with BoN inference in code generation tasks.

**Weaknesses:**

- This continuous max@k optimization method is a direct extension of the previous works on binary reward setting. The novelty is limited.
- The empirical results mainly focus on code generation tasks, and with only the Qwen2.5-coder-7B model. It is hard to conclude that the proposed method can generalize to other tasks and models.
- The max@1 score compared with the baselines is not significantly improved which raises the question of whether the proposed method can improve the single-sample accuracy.
- The results only demonstrate the effectiveness of the off-policy BoN objective in max@k while k=128. It lacks the understanding of the performance with different k values.

**Questions:**

- The experiments are limited to code generation tasks using only the Qwen2.5-Coder-7B model. Can you provide results on other domains (e.g., mathematical reasoning) or with different model architectures to validate its generalization？
- The improvement on the max@1 metric compared to baselines is not significant. Does your method inherently trade off single-sample accuracy for better BoN performance, or can it be tuned to improve both?
- The experiments focus on k=128 for evaluating the off-policy BoN objective. Could you provide results or analysis for different values of k (e.g., 256, 512) to better understand how the method scales with the number of samples?

---

> ### Author Response · Authors · 2025-11-20
> **Response to Reviewer THfK**
>
> Thank you for taking the time to review our paper and for providing valuable feedback. We have revised the manuscript accordingly and address your specific comments below.
>
> >The experiments are limited to code generation tasks using only the Qwen2.5-Coder-7B model. Can you provide results on other domains (e.g., mathematical reasoning) or with different model architectures to validate its generalization
>
> Regarding other domains, we have not conducted experiments because our approach relies on continuous rewards, and we are not aware of verifiable continuous rewards for mathematical reasoning. As for other model architectures, we were able to run additional experiments with [ByteDance-Seed/Seed-Coder-8B-Instruct](https://huggingface.co/ByteDance-Seed/Seed-Coder-8B-Instruct). On four datasets out of five our modification of on-policy bon estimate (bon mean) and our off-policy estimation (bon off) are top 2 performing approaches. On livebench we are not sure what happened with the highest score, but we are investigating it.
>
> | | CodeContests | LiveCodeBench | LiveBench | CodeForces | MBPP |
> |---|---|---|---|---|---|
> | bon-max second | 0.707 | 0.850 | 0.318 | 0.280 | 0.733 |
> | bon-max mean | 0.708 | 0.847 | 0.446 | 0.298 | 0.725 |
> | bon loo-1 | 0.699 | 0.797 | 0.338 | 0.204 | 0.737 |
> | bon mean | 0.745 | 0.861 | 0.740 | 0.344 | 0.746 |
> | bon off | 0.738 | 0.864 | 0.429 | 0.451 | 0.761 |
>
> >The improvement on the max@1 metric compared to baselines is not significant. Does your method inherently trade off single-sample accuracy for better BoN performance, or can it be tuned to improve both?
>
> Neither our method nor the baseline methods explicitly target single-sample accuracy; the comparison is included to show that our training does not degrade max@1 performance relative to the base model. There do exist techniques to optimize for both diversity and single-sample accuracy, for example, approaches that optimize Best-of-1 distribution (standard single-shot generation) to match Best-of-N distribution  [(Sessa et al., 2024)](https://arxiv.org/abs/2407.14622), [(Amini et al., 2024)](https://arxiv.org/abs/2407.06057), and [(Gui et al. 2024)](https://arxiv.org/abs/2406.00832) or approaches that decrease k value during training from high values to 1  [(Walder & Karkhanis 2025)](https://arxiv.org/abs/2505.15201) and [(Chen et al. 2025)](https://arxiv.org/abs/2508.10751). You can find more information about them in section 2.3. However, based on our results, our method does improve max@1 performance compared to the base model, therefore, exploring mentioned approaches is beyond the scope of our current work.
>
> > The experiments focus on k=128 for evaluating the off-policy BoN objective. Could you provide results or analysis for different values of k (e.g., 256, 512) to better understand how the method scales with the number of samples?
>
> Additional experiment results including k=256 and intermediate values can be found in appendix H of the paper. max@1 metric improves compared to the base model and is comparable with other baselines. max@256 is better for our approach across all datasets, but the variance of max@256 estimation is higher due to the amount of samples used in it.

---

### Official Review · Reviewer_5AKJ · 2025-11-03

**Soundness:** 3
**Presentation:** 2
**Contribution:** 3
**Rating:** 6
**Confidence:** 2

**Summary:**

Prior work derived an on-policy gradient estimator for max@k. This work derives an off-policy version, and gives experimental evidence that this can improve max@1 scores (and can remain competitive for max@k) across five RL with Verifiable Rewards (RLVR) coding benchmarks.

**Strengths:**

Simple

Clear with few exceptions (noted in weaknesses)

I am not familiar with the related works, but I believe the off-policy gradient estimator is novel

Four reasonable baseline algorithms in main results

Includes both code snippets in the paper, and full code in the supplementary materials

**Weaknesses:**

The abstract says "We derive an unbiased on-policy gradient estimate for direct optimization of this metric." But my understanding is that exact (on-policy) estimator was already derived in prior work (as noted at the start of section 3.2).

The bulleted list of contributions at the end of Section 1 also repeats that claim, but again I think it maybe shouldn't.

That same list of contributions includes "We show that optimization of continuous reward is crucial for successful RLVR application", but I think that claim should be worded a lot less strongly.

A big weakness is the `clamp_delta` hyperparameter. It's great that the code snippet is included, but it's bad that this hyperparameter is not mentioned anywhere else in the paper, and it also seems a potentially big downside of the algorithm.

Doesn't explain why those particular hyperparameters in Section 4.4 were chosen. I searched quickly just now, and a learning rate of 5e-6 is a common default, and it looks like the other hyperparameters might likewise be common defaults, but I think this should probably be mentioned in the paper.

**Questions:**

> Yue et al. (2025) investigate the effect of RLVR finetuning on math, coding, and visual reasoning benchmarks. They show that while RLVR improves pass@1 performance, the base model achieves higher pass@k scores for larger k.

If I understand this right, Yue et al. find more intuitive results for pass@1 than the maybe counterintuitive binary reward results found in this submission's Table 1? Why might that be? It might be helpful to note this contrast.

&nbsp;

> The policy gradient family of RL algorithms became standard in LLM post-training. Introduced by Stiennon et al. (2020), the key idea of these approaches is to consider the language model as a policy and train it with an RL algorithm.

Would it be fair to say Ranzato et al. ([https://arxiv.org/abs/1511.06732](https://arxiv.org/abs/1511.06732)) or Ziegler et al. ([https://arxiv.org/abs/1909.08593](https://arxiv.org/abs/1909.08593)) introduced policy gradient training of LLMs, rather than Stiennon et al.?

&nbsp;

> However, BoN can be too computationally expensive, and several works addressed this issue. A range of works address this limitation by fine-tuning the model to directly mimic the BoN distribution: [...]. This line of work is orthogonal to the present study since their inference strategy assumes generation of a single completion, as opposed to the generation of k completions employed here.

Just how orthogonal are they? Could any of those works be modified to generate $k$ completions in an optimized way?

&nbsp;

> Another line of work explores model’s optimization to maximize performance with an account for inference-time techniques, like BoN.

A bit ungrammatical

&nbsp;

> Applying the log-derivative trick

The $\theta$ after this part runs into $\pi$, it needs better spacing

A similar issue happens for $\theta$ and $\rho$, later, in Section 3.3, after "Similarly to on-policy scenario, we can calculate gradient of the objective"

&nbsp;

> All other deltas appear in the sum $\binom{i-3}{k-3}$

Should that be $\binom{j-3}{k-3}$?

&nbsp;

> Adam optimizer(Kingma & Ba, 2014)

missing space

&nbsp;

> with learning rate $5e – 6$

As rendered in the paper, the 5e-6 part looks like 5 times Euler's number minus 6

&nbsp;

$k$ vs k (math font vs plain text font) are used inconsistently to mean the same thing (throughout the paper)

&nbsp;

---

> ### Author Response · Authors · 2025-11-20
> **Response to Reviewer 5AKJ**
>
> We are really thankful for your time going through the whole paper! Your feedback is really valuable to us and we updated our paper accordingly to all of your comments. Below we will address some of your other concerns that did not make it to the paper.
>
> >Yue et al. (2025) investigate the effect of RLVR finetuning on math, coding, and visual reasoning benchmarks. They show that while RLVR improves pass@1 performance, the base model achieves higher pass@k scores for larger k. If I understand this right, Yue et al. find more intuitive results for pass@1 than the maybe counterintuitive binary reward results found in this submission's Table 1? Why might that be? It might be helpful to note this contrast.
>
> The difference in results between our work and Yue's work primarily comes from the number of test cases per problem. Yue et al. trained on the [ganler/code-r1-12k](https://huggingface.co/datasets/ganler/code-r1-12k) dataset, which contains on average 2.5 tests per problem (median 2), whereas our experiments used the CodeContests dataset, with an average of 99 tests (median 100). In our preliminary experiments on a smaller dataset (MBPP, with 3 tests per problem), we also observed a difference between binary and continuous rewards, though it was much less pronounced. The reason for such behaviour is when there are few test cases, binary and continuous rewards behave similarly, but as the number of tests increases, the binary reward becomes too sparse, making optimization harder and highlighting the advantage of continuous rewards.
>
> > However, BoN can be too computationally expensive, and several works addressed this issue. A range of works address this limitation by fine-tuning the model to directly mimic the BoN distribution: [...]. This line of work is orthogonal to the present study since their inference strategy assumes generation of a single completion, as opposed to the generation of k completions employed here. Just how orthogonal are they? Could any of those works be modified to generate  completions in an optimized way?
>
> The approaches mentioned in this section aim to align the distribution of a single generation from the fine-tuned model with the BoN distribution of the initial model. By “orthogonal,” we mean that these methods can be applied independently of our work or on top of it. To address the question, any of the approaches cited could be applied to a model trained with our method (or other methods) to further improve pass@1 performance.

---

### Official Review · Reviewer_A8Cg · 2025-11-06

**Soundness:** 2
**Presentation:** 3
**Contribution:** 2
**Rating:** 4
**Confidence:** 2

**Summary:**

This paper investigates fine tuning Large Language Models (LLM) with Reinforcement Learning to perform well on coding tasks. In particular, the paper focuses on best of N sampling at inference time, where N solutions are generated and a verifier selects the best performing solution to be returned.

The authors show that fine tuning an LLM with a policy gradient algorithm (GRPO) reduces the diversity of its generations, where performance of a single generation improves. But if we generate N solutions and evaluate the best one, fine tuning with RL hurts performance.

The paper uses a continuous reward function defined as the ratio of unit tests passed on a generated program by the LLM instead of a binary signal to check if all tests have passed. This leads to the max@K objective as the expected value of highest reward among the generations. The authors continue to derive an estimate of this objective in on-policy and off-policy settings, to be used with GRPO. Finally, experiments on 5 coding domains show that fine tuning an LLM with the proposed algorithm outperforms other baselines in 4 out of 5 domains under max@K scheme but does not improve performance under max@1 scheme.

**Strengths:**

1. The paper’s writing and organization is clear and coherent, I could mostly follow the motivation for the research and how this work fits with the covered prior work.
2. The experiment section contains sufficient details regarding the methodology and evaluation scheme, the choice of hyperparameters, baselines, and statistical tests performed to evaluate the significance of the results.
3. The derivations of the GRPO objective adapted to max@K in section 3 is clear and easy to follow.

(I should point out that I am not an LLM expert, and therefore I do not say anything strong regarding the novelty or significance of this work to the LLM community but do raise a concern regarding the applicability of best of N sampling in the Weaknesses section.)

**Weaknesses:**

1. The authors interchangeably use K and N in the text. This should be unified and cleaned up.
2. I did not understand the first experiment (Table 1). The authors claim optimization of continuous reward function is crucial for fine tuning LLMs with RL. While this intuitively makes sense to me as an RL expert, I missed how this experiment suggests such a claim.
3. Perhaps the biggest shortcoming of this work, in my opinion, is conceptual:
  - Optimizing max@K performance of the model does improve its performance under best of N sampling, since the optimization objective better matches the evaluation scheme but it may also increase the variance of generations making each output less reliable.
 - While this issue may not show up in benchmark performance under best of N sampling, it does raise concerns for the applicability of such model in the real world. Not every coding problem in the world comes with a suite of verifiable tests. And the additional cost of computation and verification would limit the usability of such a model.
- Optimizing the performance of best of N generations changes the distribution of outputs such that given sufficiently large N, a very good solution is generated. This is an effort to produce a large set of generations that contain a good answer (like finding a needle in a haystack), rather than producing reliable confident answers (planning and reasoning to complete a difficult task).
- It would be good to include a discussion regarding the variance of generations and the distribution of quality of generated solutions to see whether optimizing the max may harm the performance of other solutions.

**Questions:**

Please respond to the raised issues in points (2) and (3) of the weaknesses section of this review.

---

> ### Author Response · Authors · 2025-11-20
> **Response to Reviewer A8Cg**
>
> Thank you for your time reviewing our paper and your valuable comments. We updated our paper accordingly and below will go through some particular questions.
>
> >I did not understand the first experiment (Table 1). The authors claim optimization of continuous reward function is crucial for fine tuning LLMs with RL. While this intuitively makes sense to me as an RL expert, I missed how this experiment suggests such a claim.
>
> In Table 1, we compare two types of reward functions used during training: a continuous reward $r\_c$ and a binary reward $r\_b=[r\_c==1]$. We train two separate models—one with $r\_c$​ and the other with $r\_b$​—while keeping all other settings identical. The results in Table 1 show that training with the continuous reward improves the base model’s performance on pass@1, while slightly decreasing pass@128. In contrast, training with the binary reward leads to performance degradation across all metrics. This demonstrates that preserving the reward’s continuous structure provides more informative feedback during optimization, leading to better fine-tuning outcomes. Moreover, Evaluation of pass@k metric is based on the binary reward $r\_b$. That is why it is expected for pass@k to increase after RLVR with $r\_b$ as a reward. However, Table 1 shows that this is not true (pass@1 drops from 0.211 to 0.092 after RLVR with binary reward). But with the continuous reward, the results of RLVR are as in [(Yue et al., 2025)](https://arxiv.org/abs/2407.14622) (pass@1 raises, pass@128 drops). This result also aligns with [(Razin et al. 2025)](https://arxiv.org/abs/2503.15477) work, where they show that variance in the reward values is also very important for optimization. That motivates our choice to proceed with continuous reward instead of binary reward for further experiments.
>
> > the optimization objective better matches the evaluation scheme
>
> To avoid this mistake, we are reporting pass@k for different k. While the optimization objective targets higher k, we still got significant gains for lower k (including k=1).
>
> > This is an effort to produce a large set of generations that contain a good answer (like finding a needle in a haystack), rather than producing reliable confident answers (planning and reasoning to complete a difficult task).
>
> There exist methods to maintain the quality of individual generations while also improving best-of-N performance, such as approaches optimizes Best-of-1 distribution (standard single-shot generation) to match Best-of-N distribution, for instance, [(Sessa et al., 2024)](https://arxiv.org/abs/2407.14622), [(Amini et al., 2024)](https://arxiv.org/abs/2407.06057), and [(Gui et al. 2024)](https://arxiv.org/abs/2406.00832). Another approach applicable to our method is training with decreasing k values, this approach shown to be efficient for improving both single-shot and mult-shot generations by  [(Walder & Karkhanis 2025)](https://arxiv.org/abs/2505.15201) and [(Chen et al. 2025)](https://arxiv.org/abs/2508.10751). However, in our experiments, we did not observe any drop in single-generation performance across all datasets, as shown in Table 3 and Appendix H. For example, Table 5 shows that on the CodeContest dataset our method improves pass@1 by +3.6 p.p., compared with +5.0 p.p. for the best baseline; for pass@2, the gains are +2.9 p.p. (ours) versus +3.9 p.p. (best); and for pass@128, our method achieves +1.2 p.p. while the best baseline actually drops to –1.3 p.p. We observe a similar pattern on all the reported datasets.  Therefore, applying techniques specifically to preserve max@1 performance was not necessary and was outside the scope of this work.
>
> > Not every coding problem in the world comes with a suite of verifiable tests.
>
> That is true, that is why we always report pass@1/max@1 to demonstrate that any single answer is still reliable. Yet, in some cases the test suite is available: in test-driven development, when tests act as a specification, or in math domain where tests can describe the problem.
>
> > It would be good to include a discussion regarding the variance of generations and the distribution of quality of generated solutions to see whether optimizing the max may harm the performance of other solutions.
>
> We used max@1 metric as a check for degradation in one-shot generation quality. If the variance induced by max@k optimization were harmful for individual solutions, we would see a decline in the max@1 metric. However, the results from Table 3 depict the opposite case.

---

### Author Response · Authors · 2025-11-20
**General comment**

We sincerely thank the reviewers for their thorough evaluation and feedback. All concerns have been carefully considered, and corresponding changes have been made to the paper. Detailed responses to each reviewer's questions are provided below. We believe these revisions address the feedback and strengthen the contribution without altering the method or conclusions.

Additionally we want to address concerns about the novelty of our work. While [Walder & Karkhanis 2025](https://arxiv.org/abs/2505.15201) provide derivation for pass@k (binary reward) and max@k (continuous reward) objectives, they don’t provide any real data experiments with it and report only with pass@k results. In our work, we show that their exact approach (with loo-1 regularisation) is suboptimal and doesn’t improve max@k for large k compared to the base model (table 2 row 1 vs row 4). Additionally, we show that a change to the variance reduction can improve its performance (bon mean in table 2). And on top of that, switching from on-policy estimate to off-policy estimate improves it further. These results highlight meaningful extensions beyond prior work.

---

### Meta-Review · Area_Chair_jUo4 · 2026-01-06

**Summary:**

The manuscript considers reinforcement learning with verifiable rewards and proposed a max@k metric to address the degradation of diversity in best-of-N sampling approaches. While the empirical result seems promising, the reviewers feel that it lacks significant novelty, which the meta-reviewer tends to agree after reading the manuscript.

**Reviewer Concerns:**

The authors addressed most of the concerns raised by the reviewers.

**Reviewer Scores:**

I believe the reviewers will keep their scores.

---

### Decision · Program_Chairs · 2026-01-26

Reject